# Ocular Adverse Events after Inactivated COVID-19 Vaccination

**DOI:** 10.3390/vaccines10060918

**Published:** 2022-06-09

**Authors:** Zhihua Li, Feng Hu, Qian Li, Shuang Wang, Chunli Chen, Yongpeng Zhang, Yu Mao, Xuehui Shi, Haiying Zhou, Xusheng Cao, Xiaoyan Peng

**Affiliations:** 1Beijing Tongren Eye Center, Beijing Tongren Hospital, Capital Medical University, 17 Hougou Lane, Chongnei Street, Beijing 100005, China; lizhihua919@sina.com (Z.L.); hfe00116023@163.com (F.H.); liqian405@126.com (Q.L.); wangshuangophthal@163.com (S.W.); chenchunli125@163.com (C.C.); havenotzhang@163.com (Y.Z.); canghai19841119@sohu.com (Y.M.); shixuehui212@126.com (X.S.); ying-3114@163.com (H.Z.); 13520023405@163.com (X.C.); 2Beijing Institute of Ophthalmology, Beijing 100005, China; 3Beijing Ophthalmology and Visual Science Key Laboratory, Beijing 100005, China

**Keywords:** COVID-19, inactivated vaccine, adverse event, ocular disorder, uveitis, retinal vascular occlusion, retinal hemorrhage

## Abstract

Purpose: To report the clinical characteristics of ocular adverse events that have occurred, in China, after vaccination with inactivated COVID-19 vaccines. Methods: A retrospective cross-sectional observational study was conducted of ocular disorders that occurred within 15 days from any dose of an inactivated COVID-19 vaccine. Information on gender, age, the interval between the vaccination and ocular symptoms, laterality, duration of the ocular symptoms, primary visual acuity, and clinical diagnosis were retrospectively collected. Results: Twenty-four patients were involved in the study, including 15 females and 9 males, with a mean age of 41 ± 16 years (range of 8–71 years). The patients all denied a prior history of COVID-19 infection. Ocular adverse events occurred after the first dose of vaccine in 18 patients and, after the second or third doses, in six patients. The interval between vaccination with the inactivated COVID-19 vaccine and ocular symptoms was 6 ± 5 days; six patients were bilaterally involved and 18 patients were unilaterally involved. Regarding the diagnosis, 10 patients were diagnosed with white dot syndrome (WDS), 9 patients were diagnosed with uveitis, and 5 patients were diagnosed with retinal vascular disorders. The ages of patients with WDS were younger than those with uveitis or retinal vascular disorders (32 ± 10 vs. 48 ± 18, *p* < 0.05). For patients diagnosed with WDS, the best-corrected visual acuity (BCVA) was 0.74 ± 0.73 LogMAR. For patients diagnosed with retinal vascular disorders or uveitis, the BCVA was 1.44 ± 1.26 LogMAR. There was no significant difference (*p* > 0.05). Conclusions: A relationship cannot be established between inactivated COVID-19 vaccines and ocular disorders; therefore, further investigation of the clinical spectrum of ocular adverse events after vaccination with an inactivated COVID-19 vaccine is necessary.

## 1. Introduction

The on-going COVID-19 pandemic caused by the severe acute respiratory syndrome coronavirus (SARS-CoV-2) has led to high morbidity and mortality worldwide [1]. Effective vaccines are key to stopping the COVID-19 pandemic. Multiple vaccines against SARS-CoV-2 have been developed worldwide, including inactivated vaccines, live virus vaccines, recombinant protein vaccines, vector vaccines, and DNA or RNA vaccines [2]. Inactivated COVID-19 vaccines have been widely used among the Chinese population and have shown good safety, tolerability, and immunogenicity [3,4]. Nevertheless, ocular adverse events have rarely been reported after COVID-19 mRNA vaccines, vector vaccines, and inactivated vaccines [5,6,7]. A relationship has not be established between reported ocular disorders and COVID-19 vaccines, because of limited awareness of the clinical spectrum of potential ocular adverse events, and unclear mechanisms for ocular disorders and COVID-19 vaccines. Our study was designed to investigate the clinical picture of ocular disorders that occurred within two weeks after vaccination with an inactivated COVID-19 vaccine. Unlike a report by Dr Xiuju Chen et al., in which all the patients showed transient ocular disturbance and responded well to steroids [8], in our study, several patients had severe infectious ocular disorders or retinal vascular disorders and significantly impaired vision.

### Material and Methods

This study was a retrospective observational case series of patients who subjectively reported ocular adverse events after vaccination with an inactivated COVID-19 vaccine, and presented at the retina and uveitis service center in the Beijing Tongren Hospital (China). The study was performed in accordance with the principles of the Declaration of Helsinki. The Ethics Committee of the Beijing Tongren Hospital approved the research. Written informed consent was waived for all participants, since the study design included a retrospective recruitment of the participants and the data had been obtained during routine care of the patients.

The main inclusion criterion was the development of ocular symptoms within 15 days from any dose of an inactivated COVID-19 vaccine. The demographic information included gender and age, which was retrospectively collected. The interval between the vaccination and ocular symptoms, and doses (first, second, or third) of inactivated COVID-19 vaccine were collected. The clinical presentations included laterality of the ocular disorders, duration of the ocular symptoms, initial best-corrected visual acuity (BCVA), and clinical diagnosis, which were retrospectively collected.

The BCVA values of the patients were converted to logMAR values for further analysis. For a BCVA value below 0.01, the following LogMAR values were assigned: counting fingers vision was assigned 2.6 LogMAR, hand movement vision was assigned 2.9 LogMAR, light perception vision was assigned 3.1 LogMAR, and no light perception vision was assigned 3.4 LogMAR [9].

Patients were divided into two subgroups according to clinical diagnosis: (1) white dot syndrome (WDS) and (2) other ocular disorders (uveitis or retinal vascular disorders). Age, primary BCVA, and the interval between ocular symptoms and COVID-19 vaccination between patients diagnosed with WDS and those with retinal vascular disorders/uveitis were compared by *t*-test. Statistical significance was defined as a 2-tailed *p*-value of <0.05. The analyses were performed using the SPSS, version 17.0 software (SPSS Inc., Chicago, IL, USA).

## 2. Results

The detailed demographic and clinical information are described in Table 1. Twenty-four patients were involved in the study, including 15 females and 9 males. The ages of patients ranged from 8 to 71 years, and the mean age was 41 ± 16 years. Among the twenty-four patients with ocular adverse events, six patients were bilaterally involved and eighteen patients were unilaterally involved.

### 2.1. Diagnosis

Regarding clinical diagnosis, ten patients were diagnosed with white dot syndrome (WDS), nine patients were diagnosed with uveitis, and five patients were diagnosed with retinal vascular disorders. Among the ten patients who were diagnosed with WDS, eight patients were diagnosed with multiple evanescent white dot spot syndrome (MEWDS), one patient was diagnosed with punctate inner choroidopathy (PIC), and one patient was diagnosed with acute posterior multifocal placoid pigment epitheliopathy (APMPPE). Among the nine patients who were diagnosed with uveitis, four patients were diagnosed with non-infectious uveitis and five patients were diagnosed with infectious uveitis. Among the five patients who were diagnosed with retinal vascular disorders, four patients were diagnosed with retinal vascular occlusion and one patient was diagnosed with vitreous hemorrhage.

Among the five patients who were diagnosed with infectious uveitis, four patients were diagnosed with acute retinal necrosis (ARN), and one patient was diagnosed with ocular toxoplasmosis. For patients diagnosed with WDS, the mean age was 32 ± 10 years and, for patients diagnosed with retinal vascular disorders or uveitis, the mean age was 48 ± 18 years; there was significant difference (*p* < 0.05). For patients diagnosed with MEWDS, the mean age was 33 ± 10 years. For patients diagnosed with retinal vascular occlusion, the mean age was 53 ± 16 years. For patients diagnosed with ARN, the mean age was 56 ± 9 years.

### 2.2. Etiology

Regarding the etiology of the diseases, six patients had infectious ocular disorders, and eighteen patients had non-infectious ocular disorders. Among the six patients with infectious ocular disorders, in five patients, it was caused by herpetic virus infections and, in one patient, it was caused by a toxoplasma infection. For the four patients diagnosed with ARN, the varicella zoster virus (VZV) was detected using a polymerase chain reaction test for aqueous humor. For patient No. 14, the diagnosis of herpetic optic neuritis was made based on the typical clinical features of optic neuritis which occurred one week after the appearance of a typical facial zoster. For patient No. 24, the diagnosis of ocular toxoplasmosis was based on a typical active isolated retina-choroidal lesion, and resolution of the active lesion as well as inflammation in the vitreous body after oral compound sulfamethoxazole. Among the eighteen patients with non-infectious ocular disorders, ten patients had WDS, four patients had retinal vascular disorders, and four patients had noninfectious uveitis.

### 2.3. Anatomy

The retina, choroid, retinal vessels, optic nerve, vitreous body, and anterior chamber could be involved in ocular adverse events after vaccination with an inactivated COVID-19 vaccine. Most patients had posterior segment involvement because they were reported from the department of retina and uveitis. Patients diagnosed with WDS had outer retina and choroid involvement. Patients diagnosed with retinal vascular disorders had vessel involvements including the central and branch of the retinal artery or vein. Patients diagnosed with uveitis had anterior chamber, vitreous body, retina, and choroid involvements.

### 2.4. Visual Function

The BCVA values ranged from no light perception to 1.0. For patients diagnosed with WDS, the BCVA values ranged from counting fingers to 1.0 (LogMAR ranging from 2.6 to 0), and the mean BCVA was 0.74 ± 0.73. For patients diagnosed with retinal vascular disorders or uveitis, the BCVA values ranged from no light perception to 1.0 (LogMAR ranging from 3.4 to 0), and the mean BCVA was 1.44 ± 1.26. There was no significant difference between the BCVA values of patients diagnosed with WDS and retinal vascular disorders or uveitis (*p* > 0.05).

### 2.5. Dose and Interval

Ocular adverse events occurred after the first dose of inactivated COVID-19 vaccine in eighteen patients, after the second dose in two patients (patients No. 1 and No. 7), and after the third dose in three patients (patients No. 5, No. 12, and No. 17). For patient No. 19, the patient reported that visual acuity in the left eye declined one week after the first dose of inactivated COVID-19 vaccine, and visual acuity in the right eye declined 3 days after the second dose of inactivated COVID-19 vaccine. The intervals between the COVID-19 vaccination and ocular symptoms ranged from 1 to 15 days, and the mean interval was 6 ± 5 days. Nineteen patients had ocular symptoms within one week after the inactivated COVID-19 vaccine. For patients diagnosed with WDS, the mean interval between ocular symptom onset and vaccination with the inactivated COVID-19 vaccine was 6 ± 5 days; and for those diagnosed with retinal vascular disorders and uveitis, the mean interval was 5 ± 5 days. There was no significant difference for the interval between the ocular symptom and COVID-19 vaccination among the patients diagnosed with WDS or retinal vascular disorders and uveitis (*p* > 0.05). Administration of the standard three doses of inactivated COVID-19 vaccine was planned for all the patients; however, for patients with ocular disorders after the first or second dose of vaccine, further vaccination was halted.

### 2.6. Medical History

All the patients denied prior history of COVID-19 infection. Among five patients diagnosed with retinal vascular disorders, three patients denied prior history of systemic hypertension and diabetes (patients No. 11, No. 12, and No. 14), two patients had diabetic retinopathy and finished pan-retinal photocoagulation (patients No. 13 and No. 15). The other 22 patients denied ocular disorders prior to the vaccine administration.

## 3. Discussion

We analyzed 24 cases of ocular adverse events after vaccination with an inactivated COVID-19 vaccine, which occurred within two weeks after the vaccination and were subjectively reported by the patients. WDS was the most common ocular disorder that occurred after vaccination with the COVID-19 vaccine, in which MEWDS was the majority type. The majority of patients were unilaterally involved. The mean interval between ocular symptoms onset and COVID-19 vaccination was 6 days. Patients who developed WDS were younger than those with retinal vascular disorders or uveitis. Ocular infections including ARN, ocular toxoplasmosis, and herpetic neuritis occurred occasionally after vaccination with an inactivated COVID-19 vaccine. The majority of ocular adverse events were noted after the first dose of inactivated COVID-19 vaccine (18 out of 24 events), and only a minority of events were noted after the second or third dose. For those reporting ocular adverse events after the second or third dose, we assumed that potential ocular involvement might be mild and asymptomatic after the first dose.

In the present study, all the patients received inactivated COVID-19 vaccines (CoronaVac or BBIBP-CorV). CoronaVac (Sinovac Life Sciences, Beijing, China) is created from African green monkey kidney cells (Vero cells) that have been inoculated with SARS-CoV-2. At the end of the incubation period, the virus is harvested, inactivated with β-propiolactone, concentrated, purified, and finally absorbed onto aluminum hydroxide [3]. For the BBIBP-CorV vaccine (Sinopharm/Beijing Institute of Biological Products), the strain 19nCoV-CDC-Tan-HB02 is purified and passaged in Vero cells to generate the stock for vaccine production. To inactivate virus production, β-propionolactone is thoroughly mixed with the harvested viral solution. The vaccine is manufactured as a liquid formulation with aluminum hydroxide adjuvant (0.45 mg/mL) [10].

Sex differences in the immunological responses and adverse reactions to vaccines have been well reported. Antibody responses have been reported to be higher in females than in males following vaccinations against multiple viruses [11]. Adverse events are more common in adult females than in males for a number of vaccines [12]. Approximately 300 cases of vaccine-associated intraocular inflammation/uveitis have been reported in the literature. Vaccine-associated uveitis has been reported to show gender predominance to females (72.1%) [13]. In the present study, 15 of 24 patients with ocular disorders after vaccination with an inactivated COVID-19 vaccine were females (62.5%). Multiple factors may be helpful for explaining the sex differences in efficacy and adverse events following vaccination: (1) Among adults, females typically develop higher inflammatory, antibody, and cell-mediated immune responses to vaccines than males. (2) Sex hormones, including testosterone, estradiol, and progesterone, modulate the functioning of immune cells, including B cells, resulting in differential immune responses between the sexes. (3) Genetic differences between the sexes, including the expression of X-linked genes that may escape X-inactivation, the expression of miRNAs, and polymorphisms in immune response genes, can impact sex-differential immune responses to vaccines. (4) Gender differences in the access to and acceptance of vaccines can contribute significantly to the differential efficacy of vaccines between males and females. (5) Other factors that might be influenced by gender difference, such as diet, the microbiome, and chronic infections by viruses and parasites, might also have influences on the efficacy and adverse events following vaccination [14].

Recently, ocular adverse events have been reported to occur after vaccination with inactivated COVID-19 vaccines, showing a variable clinical spectrum. A retrospective consecutive case series including seven patients about ocular adverse events after the first dose of an inactivated COVID-19 vaccine showed variable clinical presentations including episcleritis, anterior scleritis, acute macular neuro-retinopathy, paracentral acute middle maculopathy, and subretinal fluid [6]. Li et al. reported seven patients with ocular adverse events after vaccination with an inactivated COVID-19 vaccine, in which three patients were diagnosed with Vogt–Koyanagi–Harada (VKH)-like uveitis, one patient was dianosed with multifocal choroiditis, one patient was diagnosed with episcleritis, one patient was diagnosed with iritis, and one patient was diagnosed with acute idiopathic maculopathy [8]. The prognoses of patients reported in the above two studies were good, with mild reversible clinical presentation or good response to steroids. However, in our study, we observed severe infectious ocular disorders or retinal vascular disorders that induced significantly impaired vision loss. Our study was a retrospective observational case series of patients who subjectively reported ocular adverse events after vaccination with an inactivated COVID-19 vaccine, rather than enquiring about prior recent COVID-19 vaccination history, which might partially explain the severe clinical pictures of our study.

In our study, the most common clinical diagnosis for ocular adverse events was WDS (10 out of 24 cases), in which MEWDS was the major type (8 out of 10 cases). The exact pathophysiology of MEWDS is not clear. It is assumed that it is due to an immune-mediated mechanism manifested in either the outer retina, choriocapillaris/inner choroid, or both, in genetically predisposed individuals [15]. In addition to after vaccination with an inactivated COVID-19 vaccine, MEWDS has also been reported to occur after various vaccinations, including the mRNA COVID-19 vaccine [16], as well as, influenza [15], rabies [17], human papilloma virus [18], hepatitis A and B [19,20], meningococcal [21], and yellow fever vaccines [19]. Specifically, vaccines have been suggested to trigger an inflammatory response resulting in uveitis by means of molecular mimicry or direct antigen-mediated cellular/humoral immune response, or adjuvant-mediated inflammation [22,23]. Patients diagnosed with MEWDS in the present study was 33 years old, which was comparable to that of patients diagnosed with MEWDS not associated with vaccination (27 years old) [24].

In our study, retinal vascular disorders were observed after vaccination with an inactivated COVID-19 vaccine in five patients, including four patients with retinal vascular occlusion and one patient with vitreous hemorrhage. The interval was 1 day for one patient diagnosed with branch retinal arterial occlusion, and shorter than 1 day for two patients diagnosed with central retinal arterial occlusion and vitreous hemorrhage, respectively. Retinal arterial occlusion has been previously reported after vaccination with the mRNA COVID-19 vaccine, and were characterized by non-embolic, not accompanied by carotid plague or stenosis, rapid development within 2 weeks after vaccination, and without marked abnormalities of laboratory tests (such as increased D-dimer and C-reactive protein) [7]. The mean age of patients diagnosed with retinal vascular occlusion in the present study was 53 years, which was comparable to that of patients diagnosed with retinal vascular occlusion not associated with vaccination (53 years old) [25].

The mechanisms for retinal vascular disorders after vaccination with an inactivated COVID-19 vaccine are unclear, and it is also likely that the occurrences were incidental. Immune inflammation involving endothelium cells after vaccination with COVID-19 vaccines has been associated with thrombosis and leukocytoclastic vasculitis. Thrombosis after vaccination with COVID-19 vaccines (vector vaccine and mRNA vaccine) has been reported; its pathogenesis is under investigation, and currently available histopathological findings have indicated it is likely due to an underlying immune vasculitis [26]. Vasculitis has also been rarely reported after vaccination with COVID-19 vaccines (including inactivated vaccines) [27], and it has been hypothesized that an inflammatory response to the vaccine component targets the endothelium and results in small-vessel vasculitis [28]. We assumed that inflammation involving retinal vessels might be associated with the occurrence of retinal vascular disorders.

We reported six patients with infectious ocular disorders after the first dose of inactivated COVID-19 vaccine, including four cases of ARN, one case of herpetic neuritis, and one case of ocular toxoplasmosis. Herpetic eye disorders have previously been reported after mRNA and vector vaccines, and presented as conjunctivitis, stromal keratitis, and herpes zoster ophthalmicus [29]. ARN has previously been reported to occur after the mRNA COVID-19 vaccine in immunocompetent individuals [30,31], or in COVID-19 positive patients [32]. Our study demonstrated that ARN might occur after vaccinatin with an inactivated COVID-19 vaccine as well. The mean age of patients diagnosed with ARN in the present study was 56 years, which was comparable to that of those diagnosed with ARN not associated with vaccination (50 years old) [33].

The mechanism for ARN after vaccination with an inactivated COVID-19 vaccine is unclear. The development of ARN might be due to immune dysregulation after vaccination with an mRNA or inactivated COVID-19 vaccine or infection. Although ARN has mainly occurred in clinically immunocompetent individuals, a previous study revealed impaired cellular response and increased humoral response in patients with ARN [34]. Multiple mechanisms might play certain roles in the virus reactivation after vaccination, which include: reduction of neurotrophic factors, subtle immunologic deficiency, T cells drawn towards the vaccination response allowing viral reactivation unchecked, and vaccine-induced inflammatory cytokines may trigger viral activation directly or indirectly via neurotoxicity or loss of neurotrophic support [35]. The CoronaVac and BBIBP-CorV vaccines both contain aluminum salts as adjuvants. Adjuvants drive innate immune system pattern recognition receptor activation and are commonly used in vaccines to boost the immune reactivity response. Aluminum adjuvants primarily enhance antibody production, induce antibody-mediated protection by CD4+ T cells, and have little effect on the cell-mediated immune response. Several studies have suggested that the CD4+ effector cells were activated by aluminum adjuvants. The CD4+ effector cells secret multiple cytokines including interferon-γ (IFN-γ), interleukin (IL)-4, IL-5, IL-13, IL-17, and IL-22 [36]. A study about the immune status of patients with ARN revealed that serum levels of IFN-γ, IL-4, IL-5, IL-13, IL-17, and IL-22 were significantly higher in patients with ARN than in a control group. Based on these studies, we inferred that the status of activated CD4+ T cells in individuals following vaccination with aluminum adjuvants might be associated with the occurrence of ARN [37].

## 4. Limitations

The main limitation of the study was its retrospective cross-sectional design. The ages of patients diagnosed with WDS, retinal vascular occlusion, and ARN in the present study were comparable to those of patients not associated with vaccination. It is hard to exclude the incidence for occurrence of ocular disorders after the COVID-19 vaccination or establish relationships between ocular disorders and vaccination with inactivated COVID-19 vaccines. A systemic rheumatologic evaluation was not performed in our study.

## 5. Conclusions

Given the significant personal and public health benefits of COVID-19 vaccines and the rare occurrence of adverse events, we do not suggest withholding vaccination. Clinicians should be aware of the possibility of ocular adverse events including severe infectious disorders and retinal vascular disorders, and should encourage patients to pursue ophthalmic reviews if they have ocular symptoms after vaccination with an inactivated COVID-19 vaccine. The relationships between ocular disorders reported in this study and inactivated COVID-19 vaccines needs further investigation.

## Figures and Tables

**Table 1 vaccines-10-00918-t001:** Demographic and clinical information of patients with ocular adverse events after vaccination with an inactivated COVID-19 vaccine.

No.	Gender	Age	Dose ^#^	Interval *	Laterality	Diagnosis	Duration (Days)	Initial Visual Acuity
1	F	18	2	5	R	MEWDS	17	**0.9**/0.9
2	M	49	1	7	L	MEWDS	30	0.7/**0.1**
3	F	25	1	4	R	MEWDS	3	**0.6**/1.0
4	F	42	1	2	B	MEWDS	10	**FC**/**0.4**
5	M	24	3	15	L	MEWDS	14	0.8/**0.2**
6	F	38	1	15	L	MEWDS	14	1.0/**0.4**
7	F	37	2	3	R	MEWDS	7	**0.2**/0.5
8	F	31	1	2	R	MEWDS	14	NA
9	F	32	1	3	L	PIC	210	0.4/**0.05**
10	F	21	1	7	B	APMPPE	7	**0.15**/**1.0**
11	F	55	1	1	L	BRAO	30	1.0/**1.0**
12	M	33	3	7	R	BRVO	7	**1.0**/1.0
13	M	71	1	<1	R	CRAO	30	**0.02**/0.4
14	M	54	1	8	R	Herpetic optic neuritis, CRVO	5	**LP**/1.0
15	M	58	1	<1	R	Vitreous hemorrhage	7	**FC**/0.6
16	F	48	1	14	B	VKH	7	**0.3**/**0.3**
17	M	41	3	5	B	VKH	5	**0.3**/**0.2**
18	M	8	1	1	R	Posterior uveitis	30	**LP**/1.0
19	F	52	1, 2	7, 3	B	Anterior and intermediate uveitis	25/40	**0.4**/**0.4**
20	F	55	1	5	L	ARN	14	1.0/**0.1**
21	F	67	1	14	L	ARN	20	0.7/**FC**
22	F	46	1	7	R	ARN	30	**0.1**/1.0
23	F	57	1	4	B	ARN	10/20	**FC**/**NLP**
24	M	22	1	5	R	Ocular toxoplasmosis	14	**0.9**/1.2

Notes: Visual acuity in bold were from the involved eyes. **^#^**, Dose of vaccination followed by ocular disorders; *****, interval between vaccination and ocular symptoms; F, female; M, male; R, right; L, left; B, bilateral; MEWDS, multiple evanescent white dot spot syndrome; CRVO, central retinal venous occlusion; AION, anterior ischemic optic neuropathy; BRAO, branch retinal artery occlusion; BRVO, branch retinal venous occlusion; CRAO, central retinal artery occlusion; PIC, punctate inner choroidopathy; VKH, Vogt–Koyanagi–Harada syndrome; APMPPE, acute posterior multifocal placoid pigment epitheliopathy; ARN, acute retinal necrosis; CRVO, central retinal venous occlusion.

## Data Availability

All the data was included in the manuscript.

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
