# Peer review of "Ocular Adverse Events after Inactivated COVID-19 Vaccination"

_vaccines, 2022, doi:10.3390/vaccines10060918_

Round 1

Reviewer 1 Report

This report is interesting and carries significant information related to ocular adverse events following COVID-19 vaccination. Apart from minor spelling errors the paper is well written. This reviewer would like to know more about the vaccine that was used. Whether all patients received similar vaccine and what was the nature of the vaccine? 

Author Response

Reply: Thank you very much for your comments. CoronaVac (Sinovac Life Sciences, Beijing, China) and BBIBP-CorV (Sinopharm/ Beijing Institute of Biological Products) are inactivated COVID-19 vaccination in China, and were applied widely in population. All the patients received inactivated COVID-19 vaccine, and detailed information for exact type of vaccination was not available (CoronaVac or BBIBP-CorV). Nature of two common inactivated COVIDI-19 vaccination was added in the Discussion, please see page 4 line 160.

Reviewer 2 Report

1.Information on the ocular profile of the patients prior to the vaccine administration will add value to the study

2. What was the basis of the different doses administered to the patients? You mentioned that one/two patients were not in compliant, however the table showed almost half of the patients with 1 dose

3. Increasing the number of a patients will be beneficial.

4. Discussion about gender being a factor for developing ocular disorders after vaccine administration could make the study more interesting

Author Response

1.Information on the ocular profile of the patients prior to the vaccine administration will add value to the study

Reply: Thanks for suggestions. In the present study, two patients had diabetic retinopathy and finished pan-retinal photocoagulation (No.13 and No.15), other patients denied ocular disorders prior to the vaccine administration. Please see page 4 line 146.

2.What was the basis of the different doses administered to the patients? You mentioned that one/two patients were not in compliant, however the table showed almost half of the patients with 1 dose

Reply: Thank you very much for your comments. All the patients were planned administrating standard 3 doses of inactivated COVID-19 vaccines. However, for patients with ocular disorders after the first or second dose of vaccine, further vaccination was halted. Please see page 4 line 142. In table 1, the item “dose” represented the dose of vaccination followed by ocular disorders. Please see page 3 line 82.

3. Increasing the number of a patients will be beneficial.

Reply: Thanks for your comment. It is true that increasing the numbed of patients would be beneficial, especially when there was no established relationship between ocular disorders and COVID-19 vaccination.

4. Discussion about gender being a factor for developing ocular disorders after vaccine administration could make the study more interesting

Reply: Thanks for suggestions. Vaccine associated uveitis has been reported showed a gender predominance to female. We added related discussion, please see page 4 line 172.

Reviewer 3 Report

Li and Hu et al. investigated the clinical reports of ocular adverse events that occurred after the injection of an inactivated SARS-CoV-2 vaccine. Content is significant for understanding the potential adverse effects of this type of COVID-19 vaccine in the clinic. However, the authors do not show specific symptom data obtained from patients who showed adverse events after the vaccination. Unfortunately, the study appears to be descriptive and not beneficial enough for understanding the adverse effects of the inactivated virus vaccine for COVID-19. The detailed comments are listed below. 

  1. Line 36: correctly “morbidity”. 
  2. The authors study the effects of inactivated COVID-19 vaccines. Not only references, please add a brief explanation of this vaccine type, for example, how viruses were inactivated and vaccines were manufactured.
  3. Line 51 (material and Methods): Please add more details of data collection to improve the clarity of the studies.
  4. Line 102 (Etiology): The authors reported that “in 6 patients with infectious ocular disorders, 5 patients were caused by herpetic virus infection and 1 patient by Toxoplasma infection. 4 patients diagnosed with ARN. Please discuss the potential explanation for the relatively higher number of infections after the injection with the vaccine. The authors stated (lines 172-175) that their study was a retrospective observational case series of patients. Please rewrite this possible explanation to improve the clarity.

Author Response

  1. Line 36: correctly “morbidity”. 

Reply: Thanks for reminding. The “mobility” on Line 36 was correctly changed into “morbidity”.

  1. The authors study the effects of inactivated COVID-19 vaccines. Not only references, please add a brief explanation of this vaccine type, for example, how viruses were inactivated and vaccines were manufactured.

Reply: Thank you much for suggestions. We added information about the inactivated COVID-19 vaccination that was applied for patients in the present study. Please see page 4 line 164.

  1. Line 51 (material and Methods): Please add more details of data collection to improve the clarity of the studies.

Reply: Thanks for suggestions. Methods was revised in the manuscript, please see page 2 line 60.

  1. Line 102 (Etiology): The authors reported that “in 6 patients with infectious ocular disorders, 5 patients were caused by herpetic virus infection and 1 patient by Toxoplasma infection. 4 patients diagnosed with ARN. Please discuss the potential explanation for the relatively higher number of infections after the injection with the vaccine. The authors stated (lines 172-175) that their study was a retrospective observational case series of patients. Please rewrite this possible explanation to improve the clarity.

Reply: Thanks for your suggestion. We added possible explanation about virus reactivation after COVID-19 vaccination in discussion part, please see 6 page 255.

Round 2

Reviewer 3 Report

Thank you for addressing my comments. The authors addressed my previous concerns and significantly improved the manuscript's clarity. Now, I recommend the revised manuscript be published in Vaccines.